# Stable-Sketch: A Versatile Sketch for Accurate, Fast, Web-Scale Data Stream Processing

## ABSTRACT

Data stream processing plays a pivotal role in various web-related applications, including click fraud detection, anomaly identification, and recommendation systems. Accurate and fast detection of items relevant to such tasks within data streams, e.g., heavy hitters, heavy changers, and persistent items, is however non-trivial. This is due to growing streaming speeds, limited fast memory (L1 cache) available in current systems, and highly skewed item distributions encountered in practice. In effect, items of interest that are tracked only based on their features (e.g., item frequency or persistence value) are susceptible to replacement by non-relevant ones, leading to modest detection accuracy, as we reveal. In this work, we introduce the notion of *bucket stability*, which quantifies the degree of recorded item variation, and show that this is a powerful metric for identifying distinct item types. We propose Stable-Sketch, an elegant and versatile sketch that exploits multidimensional information, including item statistics and bucket stability, and adopts a stochastic approach to drive replacement decisions. We present a theoretical analysis of the error bounds of Stable-Sketch, and conduct extensive experiments to demonstrate that our solution achieves substantially higher accuracy and faster processing speeds than state-of-the-art sketches in a range of item detection tasks, even with tight memories. We further enhance Stable-Sketch's update throughput with Single Instruction Multiple Data (SIMD) instructions and implement our solution with P4, demonstrating real world deployment viability. We make the source code of Stable-Sketch publicly available on GitHub.

## KEYWORDS

sketch, data stream, bucket stability, heavy items, persistent items

**ACM Reference Format:**

Anonymous Author(s). 2023. Stable-Sketch: A Versatile Sketch for Accurate, Fast, Web-Scale Data Stream Processing. In *Proceedings of ACM Conference (WebConf'24)*. ACM, New York, NY, USA, 15 pages. https://doi.org/10.1145/nnnnnnn.nnnnnnn

## 1 INTRODUCTION

Measurements play a vital role in various web-centric areas, including user behavior analysis [1], web personalization [2], intrusion detection [3], web traffic analysis [4], etc. With the ever-increasing data rates, per-item monitoring becomes impractical as it demands

extensive memory resources to track all items of interest [5]. Consequently, approximate item processing has gained significant traction. In particular, probabilistic data structures called *sketches*, have been adopted for a range of item processing tasks, as they guarantees bounded detection errors with a limited memory footprint [6]. A sketch is usually initialized as a table with $m$ rows, each with $u$ memory entries (*buckets*), where each bucket keeps track of items (item identifiers) hashed to that bucket [7, 28].

Recently, the research community has paid much attention to three representative item processing tasks: heavy hitter detection [7, 8, 13, 14], heavy changer detection [15–17], and persistent item detection [18–20]. *Heavy hitter detection* focuses on finding items with frequency greater than a predefined value [21]; *heavy changer detection* is to find items whose frequencies vary dramatically in two adjacent time windows [22]; while *persistent item detection* aims to pick out items whose persistence (i.e., number of windows in which they appear) is larger than a given threshold [19]. In practice, analyzing web traffic with such patterns holds significant importance. For instance, tracking the frequency of a user's website visits over the course of a year can serve as an indicator of their persistence, suggesting a strong preference for the site. This data is highly valuable for web service providers as it allows them to improve user engagement and satisfaction, ultimately leading to higher platform retention and increased revenue.

### 1.1 Motivation

Although several sketch-based schemes have been proposed to handle the aforementioned detection tasks, achieving both high detection accuracy and fast update speed *simultaneously* remains challenging. Our work addresses key limitations of previous sketch designs, which mostly rely on *uni-dimensional* information (e.g., item frequency/persistence) to replace a stored item upon the arrival of a new one, and we aim to tackle several challenges, as follows:

(i) **Memory constraints:** To ensure high processing speeds, it is preferable to process items only utilizing cache memory. Contemporary CPU caches employ a hierarchical structure, categorized into L1, L2, and L3 cache levels. Among these, the L1 cache, although the smallest, are the fastest. Despite a general increase in overall cache size over the years, the capacity of the L1 cache remains constrained, typically in the kilobyte (KB) range. This is particularly notable in recent sketch-based studies [7, 29, 38], where the L1 cache size in experimental setups is usually no more than 64KB. This constraint necessitates sketches to be compact enough to fit within this space.

(ii) **High detection accuracy:** In practice, item distributions in data streams are highly skewed [24, 41] – most items own small frequency, while only a few are frequently encountered. Therefore, when the memory used by sketches is tight and hash collisions become frequent, the features of large/persistent items do not have sufficient opportunities to build significance, as small/non-persistent ones collectively appear with high frequency; as a result,

those items of interest may be mistakenly substituted by small/non-persistent ones, which harms detection accuracy.

(iii) **High update throughput:** Detection schemes should be capable of processing items swiftly to keep pace with high-speed data streams. Recent designs that utilize an external DRAM (Dynamic Random Access Memory) based data structure to record candidate items [6, 27] and handle hash collisions incur excessive memory access overheads and make it impossible to match high-speed line rates. Besides, the update operation should be further harness parallel acceleration techniques, such as SIMD instructions, to further enhance processing speeds.

(iv) **Ease of configuration:** Sketches should be straightforward to set up, without over-reliance on intricate parameter tuning. Strategies such as PIE [20] demand intricate tuning and the detection accuracy is highly sensitive to variations in parameter values, posing challenges when dealing with diverse data streams that have varying distributions.

(v) **Practical deployment:** Data stream processing schemes should be easily implementable on various hardware platforms, including but not limited to Field-Programmable Gate Arrays (FPGA) and programmable switches, which offer the highest processing speed but also present the most stringent design constraints.

These challenges motivate us to harness other statistics and devise Stable-Sketch, a new versatile sketch framework based on *multi-dimensional* features, which simultaneously achieves *high detection accuracy, memory efficiency, and processing speed*. We recognize that the state of each bucket can be leveraged to identify different item types. *If items stored in a bucket change frequently (indicating the stability of the bucket is low), that bucket more likely stores small-size items that can be discarded quickly; otherwise, it tends to track large items.* Based on this insight, our Stable-Sketch substitutes items recorded in buckets by computing replacement probabilities based on both item information and bucket stability. As a result, the larger the size of an item stored in a bucket and the higher the bucket stability, the harder it will be to replace that item with others. **This strategy also eliminates the need for complex parameter tuning and ensures easy deployment in practical scenarios.** Notice that even though recording the status of buckets in the sketch adds memory overhead, our results will reveal that this can be negligible compared with the achievable improvements in detection accuracy.

## 1.2 Contributions

To the best of our knowledge, Stable-Sketch is the first approach that utilizes the *bucket stability* feature for diverse item detection tasks, including heavy hitters, heavy changers, and persistent items. This brings the following key advantages. First, Stable-Sketch has **high memory efficiency** since it does not rely on additional data structures to hold candidate items and stops redundant hash operations once an item finds an available bucket, thus saving memory to record more items. Second, Stable-Sketch offers **fast processing speeds** – during the update process, it does not depend on pointers and reduces repetitive hash actions. During the query process, it only needs a scan of all buckets, leading to a short time of returning all heavy/persistent items. Third, Stable-Sketch attains **high detection accuracy**. We provide theoretical proofs of the error bounds

of our approach and demonstrate its superiority over state-of-the-art solutions via extensive trace-driven experiments. We further accelerate Stable Sketch's update speed with Single Instruction Multiple Data (SIMD) instructions [31]. Lastly, we prototype Stable-Sketch with P4 [59] and quantify its overhead, making the case for its **deployment in practice**. The source code of Stable-Sketch is available at [32].

## 2 PROBLEM STATEMENT

We first formalize the definitions of data stream and the item detection tasks of interest, while symbols used frequently are summarized in Appendix A.

**Data Stream:** A data stream $Q$ consists of a sequential series of items $f_1, f_2, \cdots, f_i, \cdots$. Each item $f$ owns a frequency and persistence value denoted by $V(f)$ and $P(f)$, respectively.

**Heavy Hitter Detection:** Given a data stream $Q$ with different items, a heavy hitter is identified within $Q$ whenever the frequency of that item surpasses a pre-set threshold, defined as $\theta N$, where $\theta$ is a user-defined parameter in the $(0,1)$ range and $N$ represents the total frequency of all items in $Q$.

**Heavy Changer Detection:** To detect heavy changers, we compare an item $f$'s frequency in two consecutive epochs, $E_1$ and $E_2$. Suppose the frequency of $f$ in these epochs is $q_1$ and $q_2$, respectively. If the absolute difference between $q_1$ and $q_2$ exceeds the established heavy changer threshold $\psi D$, item $f$ is classified as a heavy changer, where $D$ is the total absolute change of all items across two epochs.

**Persistent Item Detection:** A data stream composed of multiple items can be divided into $G$ equal and contiguous time windows. An item $f$'s persistence is quantified by the total number of windows in which it appears. If the persistence of an item is greater than a set threshold $\phi G$, where $\phi$ is a parameter in $(0,1]$, the item is categorized as persistent.

## 3 STABLE-SKETCH DESIGN

In this section, we first discuss the rationale behind our Stable-Sketch design, then delve into its data structure and basic operations. Afterwards, we explain how to deploy Stable-Sketch for different detection tasks, before formally analyzing its performance.

### 3.1 Rationale

Recall that sketches utilize summary data structures to record item information within a fixed number of buckets. Similar to [6, 7, 28], we initialize Stable-Sketch as a two-dimensional array with $m$ rows, in which each row contains $u$ buckets to record the values of items hashed to these buckets. Compared with existing approaches, the advancements Stable-Sketch brings are two-fold:

(i) Current schemes use a uni-dimensional feature for replacement decisions, mostly replacing items imprudently based on their frequency or persistence value, resulting in *many heavy/persistent items being erroneously evicted by non-heavy/-persistent ones*. To illustrate this problem, we resort to MV-Sketch [7], a state-of-the-art scheme for heavy hitter detection, and three CAIDA datasets [52]. More details about the scheme and datasets are in Section 5. We vary the memory size from 16KB to 256KB [38] and measure how many times non-heavy items mistakenly expel heavy items during the update process. As seen in Table 1, the number of wrong replacement events increases dramatically as the memory size decreases.

For instance, when the memory size is tight (16KB), the number of erroneous replacement events are 2,287× higher than when having a larger memory (256KB), under the CAIDA 2018 dataset. This indicates that MV-Sketch cannot provide enough protection for heavy items under constrained memory budgets.

**Table 1: Number of heavy items being wrongly replaced by non-heavy ones in MV-Sketch, applied to three datasets.**

| Trace \ Size | 16KB | 32KB | 64KB | 128KB | 256KB |
|---|---|---|---|---|---|
| CAIDA 2015 | 392,082 | 161,113 | 33,076 | 4,810 | 909 |
| CAIDA 2016 | 563,005 | 247,301 | 46,097 | 4,742 | 722 |
| CAIDA 2018 | 432,362 | 247,879 | 48,393 | 2,379 | 189 |

To tackle this problem, we explore another powerful metric that we introduce to provide more protection to potential heavy items and prevent them from being effortlessly expelled from buckets. In particular, the item distribution of real data streams is known to be highly skewed [24], while heavy items carry more data than non-heavy ones [25]. Therefore, it should take more effort to evict heavy items than non-heavy ones recorded in a bucket. Based on this observation, we harness the status of each bucket to identify the type of items recorded. Specifically, if the items stored in a bucket change frequently, meaning that the bucket has low stability, then the bucket is more likely to track some non-heavy items; otherwise, it indicates that the bucket tends to record heavy items.

To verify this feature, we use MV-Sketch to compare the stability of each bucket under different memory sizes (16KB, 32KB) and datasets. The computation of bucket stability is as follows: when a new item arrives, if the item recorded in the hashed bucket does not change, the stability of the corresponding bucket increases by 1; otherwise, the stability decreases by 1 (not less than 0). As shown in Figure 1, we find that the buckets that track heavy items own larger stability than those that track non-heavy ones. For instance, for the CAIDA 2015 trace, the average bucket stability for heavy items is 1.55× and 2.48× higher than for non-heavy ones under 16KB and 32KB memory, respectively. These results reveal that the bucket that records heavy items tends to have stronger stability since more attempts are required to replace them.

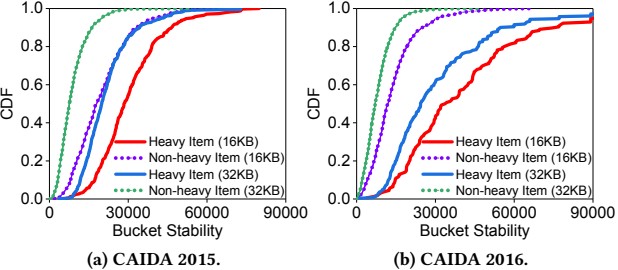

(a) CAIDA 2015.  (b) CAIDA 2016.

**Figure 1: Cumulative Distribution Function (CDF) of bucket stability for (non-)heavy items tracked under different traces and L1 caches, when employing MV-Sketch. Observe how buckets storing heavy items yield higher bucket stability.**

Therefore, our Stable-Sketch calculates a stochastic decay probability based on multi-dimensional statistics, i.e., *item information and bucket stability*, to decide whether to replace tracked items. As

recorded item statistics and bucket stability increase for a bucket, meaning that it potentially tracks a heavy item, the likelihood of this item being successfully replaced by other items decreases, thus improving detection accuracy.

(ii) Besides, recent sketch-based methods like Count-min Sketch [6] and MV-Sketch [7] hash each item in all rows, and then increment corresponding counters, which harms memory efficiency. In contrast, our Stable-Sketch, gives up repetitive hash operations once a newly arrived item finds an available bucket, to release memory for storing more items, thereby improving memory efficiency.

### 3.2 Data Structure

The data structure of recent sketches can be categorized into flat [6] and hierarchical [9, 10]. Hierarchical ones often incorporate multiple layers to enable the tracking of heavy and non-heavy items separately. Despite potential benefits in terms of accuracy, the hierarchical data structure challenges the update speed and practical deployment, especially in programmable switches with strict design constraints. Therefore, in our Stable-sketch design, we harness the conventional flat structure.

We illustrate Stable-Sketch's data structure in Figure 2, which consists of $m$ rows and $u$ columns. Each row is associated with a different pairwise-independent hash function $h_1, \cdots, h_m$. We use $B(i, j)$ to denote the bucket at the $i$-th row and the $j$-th column, where $1 \leq i \leq m$ and $1 \leq j \leq u$. Each bucket contains three fields: $B(i, j).K$ tracks the key of the current candidate item; $B(i, j).V$ stores the statistic of the candidate item, e.g., item frequency or persistence value; and $B(i, j).S$ represents the stability of this bucket. If a new item hashed into the bucket without successfully replacing the already recorded one, the stability of this bucket increases by 1; otherwise, it indicates the newly arrived item occupies this bucket, and the stability decreases by 1. Since each bucket owns a fixed memory size, the number of buckets in each row can be altered based on the pre-allocated memory size and the number of rows.

By default, Stable-Sketch keeps track of an item's key to ensure excellent invertibility. However, in certain cases, the key may be excessively long, and memory resources may be limited. In such scenarios, to further ameliorate the memory utilization of Stable-Sketch, we also propose a variant named Stable-Sketch*, in which we record the item's *fingerprint* instead of the key of the incumbent item in the bucket. The detail of Stable-Sketch* and its performance can be found in Appendix C.5. While there are techniques available for dynamically adjusting the counter size to minimize memory usage [10–12], we opt against using them. This is because they typically have a negative impact on the update and query speeds and are usually challenging to deploy on practical hardware platforms such as programmable switches.

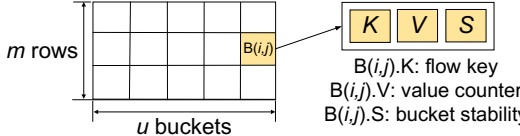

**Figure 2: Data structure of Stable-Sketch.**

### 3.3 Basic Operations

Stable-Sketch performs two main operations: 1) *Update*, which maps an arriving item into the sketch, based on multi-dimensional

information in a probabilistic manner; 2) *Query*, which returns heavy/persistent items whose estimated relevance value is greater than a specific threshold.

---

**Algorithm 1:** Stable-Sketch's Update Procedure

---

**Input:** an item $f$, hash functions $h_1, h_2, ..., h_m$, $min \leftarrow +\infty$

**1 Initialization:** The counters and item key of each bucket are initialized to 0 and *null*, respectively.

**2 for** $i = 1$ *to* $m$ **do**

**3**    **if** $B(i, h_i(f)).K == null$ **then**

**4**      $B(i, h_i(f)).K \leftarrow f.key$;

**5**      $B(i, h_i(f)).V \leftarrow 1$;

**6**      $B(i, h_i(f)).S \leftarrow 1$;

**7**      **return**;

**8**    **else if** $B(i, h_i(f)).K == f.key$ **then**

**9**      $B(i, h_i(f)).V \leftarrow B(i, h_i(f)).V + 1$;

**10**      $B(i, h_i(f)).S \leftarrow B(i, h_i(f)).S + 1$;

**11**      **return**;

**12**    **else if** $B(i, h_i(f)).V < min$ **then**

**13**      $min \leftarrow B(i, h_i(f)).V$;

**14**      $R \leftarrow i; M \leftarrow h_R(f.key)$;

**15 if** $rand(0, 1) < \frac{1}{B(R,M).V \times B(R,M).S+1}$ **then**

**16**    $B(R, M).V \leftarrow B(R, M).V - 1$;

**17**    **if** $B(R, M).V == 0$ **then**

**18**      $B(R, M).key \leftarrow f.key$;

**19**      $B(R, M).V \leftarrow 1$;

**20**      $B(R, M).S \leftarrow \max[B(R, M).S - 1, 0]$;

**21**      **return**;

**22 else**

**23**    Evict the newly arrived item;

**24**    **return**;

---

*3.3.1 Update.* Algorithm 1 gives the pseudo-code for the update process. First, all fields in the data structure are initialized to 0 or *null*. When a new item $f$ arrives, Stable-Sketch utilizes the function $h_1$ to hash $f$ to bucket $B(1, h_1(f))$. Then, one of three cases follows:

    *Case 1:* If the bucket $B(1, h_1(f))$ is empty, we insert item $f$ into this bucket and configure $B(1, h_1(f)).K$ as $f.key$, $B(1, h_1(f)).V$ and $B(1, h_1(f)).S$ as 1 (Lines 3-7).

    *Case 2:* If $B(1, h_1(f))$ has been occupied by item $f$, we increase both the value counter $B(1, h_1(f)).V$ and the stability counter $B(1, h_1(f)).S$ by 1. Otherwise, Stable-Sketch checks the buckets in the next row sequentially with the hash functions $h_2, \cdots, h_m$. Once item $f$ finds an available bucket in the $i$-th row, the hash operation terminates (Lines 8-11).

    *Case 3:* Suppose item $f$ cannot find an available bucket, indicating that it encounters hash collisions in all rows. In this case, Stable-Sketch harnesses a probability-based replacement strategy to decide whether to save or dismiss the current item $f$. Specifically, Stable-Sketch first selects the bucket with the smallest value counter among $m$ hashed buckets (Lines 12-14). Note that if multiple buckets own the same smallest value, Stable-Sketch will choose the first among them, denoted as $B(R, M)$. Then, Stable-Sketch computes a replacement probability $L(f)$ as $\frac{1}{B(R,M).V \times B(R,M).S+1}$. This

reflects that for an item saved in a bucket, the larger $B(R, M).V$ and $B(R, M).S$ are, the more challenging it will be for other items to successfully evict the stored one. If a newly arrived item fails to trigger the replacement mechanism, Stable-Sketch will discard this item. Otherwise, Stable-Sketch will decrease $B(R, M).V$ by 1. If $B(R, M).V$ reaches 0, Stable-Sketch will update the item key with that of the newly arrived one, decrease $B(R, M).S$ by 1, and set $B(R, M).V$ to 1 (Lines 15-21). Compared with probability-based replacement [34], probability-based decay ensures that the estimation error is strictly one-sided, i.e., potentially exhibiting only underestimation. An investigation of the impact of different replacement probabilities expressions $L(f)$ on the performance of the detection accuracy is available in Appendix C.4.3.

*3.3.2 Query.* For item queries, Stable-Sketch scans all buckets and if the estimated value $B(i, j).V$ of an item $f$ is greater than a predefined threshold, then $f$ is considered an item to be found.

## 3.4 Applying Stable-Sketch to Different Detection Tasks

We deploy Stable-Sketch for three applications: finding heavy hitters, heavy changers, and persistent items. Note that Stable-Sketch can be easily applied also to other tasks, such as finding superspreaders [28, 46, 47], significant items [48] and bursts [49, 50]. Due to space limitations, we do not include results for these tasks here.

*3.4.1 Heavy Hitter Detection.* Given that Stable-Sketch can be directly employed for detecting heavy hitters, the update and query processes remain consistent with what has been detailed in Section 3.3. To enhance comprehension of the update operation in Stable-Sketch, we include several illustrative examples, summarized in Figure 3. For these examples, we assume a sketch with three rows, each containing two buckets.

    *Case 1:* When item $f_5$ arrives, it uses the hash function $h_1$ to locate an available bucket in the first row. Given that the hashed bucket is currently empty, we can insert $f_5$ into the sketch and update the structure from $(Null, 0, 0)$ to $(f_5, 1, 1)$. As $f_5$ has been successfully inserted, we terminate the hash operation to conserve memory for storing other items and to reduce the update time.

    *Case 2:* When item $f_6$ arrives, it attempts to locate an available bucket by hashing in each row sequentially. Eventually, $f_6$ successfully finds a match in the third row. As a result, both the frequency counter and stability counter are incremented by one, updating the structure from $(f_6, 4, 4)$ to $(f_6, 5, 5)$.

    *Case 3:* When item $f_7$ arrives, it experiences hash collisions across all rows in the sketch. Consequently, it searches for the bucket that contains the smallest value counter ($f_4$) to initiate a decay operation on the current item. This decision is guided by the probability $\frac{1}{3 \times 3+1}$. If the decay operation succeeds and reduces the value counter to 0, item $f_7$ replaces the current item in the bucket. Otherwise, item $f_7$ is discarded.

*3.4.2 Heavy Changer Detection.* We compare two sketches at the end of two consecutive epochs $E_1$ and $E_2$ to find heavy changers. For each epoch, we construct a Stable-Sketch to record the frequency of each item and the insertion process is the same as in Section 3.3. During the query process, if the frequency difference of item $f$ is greater than the heavy changer threshold, item $f$ is viewed as a heavy changer.

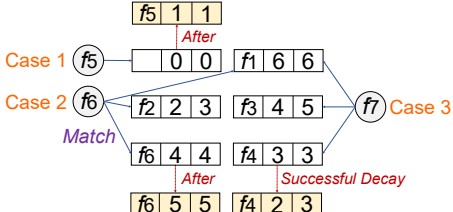

**Figure 3: Examples of the update procedure in Stable-Sketch for the heavy hitter detection task.**

*3.4.3 Persistent Item Detection.* Since the persistence of an item only increases by one at each time window, we add a flag bit (*On/Off*) in Stable-Sketch's data structure to remove duplicates [18]. Status *On* is representative of an arrived item that has not yet accessed the mapped bucket in this time window, and then Stable-Sketch increments both counters by one and turns the flag to *Off*. At the beginning of each time window, all the flags in buckets are reset to *On*. During the insertion process, Stable-Sketch first finds an available bucket for item $f$. If this fails, it will find the bucket with the smallest persistence value to conduct the replacement. If the flag bit of the selected bucket is *Off*, signifying that the saved item has arrived in this window, the newly arrived item will abandon the replacement. Otherwise, Stable-Sketch executes the replacement procedure as in Algorithm 1 (Lines 15-24). The query process for reporting persistent items is consistent with Section 3.3.

## 4 MATHEMATICAL ANALYSIS

We verify that Stable-Sketch has one-sided error and establish the error bound using heavy hitter detection. Finally, we conduct a detailed empirical study to validate our theoretical results.

### 4.1 No Over-estimation Error

THEOREM 4.1. *For any given item $f$, let $V_t(f)$ and $\hat{V}_t(f)$ denote the actual frequency and estimated frequency at a particular time $t$, respectively. Then $\hat{V}_t(f) \le V_t(f)$.*

PROOF. The detailed proof can be found in Appendix B.1.  □

### 4.2 Error Bound of Stable-Sketch

To derive the error bound, we make an assumption that is generally valid (confirmed in Appendix B.3.2): once a heavy item enters a bucket, it remains in the bucket until the detection task is complete. Then we get the error bound of Stable-Sketch as

THEOREM 4.2. *Given a small positive number $\beta$ and a heavy item $f$ with frequency $V(f)$, the inequality $\Pr\left\{V(f) - \hat{V}(f) \ge \lceil \beta N \rceil\right\} \le \frac{[\ln(V(f))+\varphi]}{\beta N \ln(S)}$ holds, where $\varphi$ denotes the Euler-Mascheroni constant, $S$ denotes the bucket stability that records item $f$, and $N$ represents the total number of entries for all items.*

PROOF. A comprehensive derivation of the bound is available in Appendix B.2.  □

### 4.3 Empirical Validation

We perform a series of empirical evaluations to validate the accuracy of Theorem 1, assess the plausibility of our assumption, verify the correctness of Theorem 2, and compare our method with existing approaches. For comprehensive information regarding the empirical validation process, please refer to Appendix B.3.

## 5 EXPERIMENTAL RESULTS

To demonstrate the performance of Stable-Sketch, we conduct experiments on a server equipped with an 8-core Intel(R) Xeon(R) W-2123 CPU @ 3.60GHz and 32GB DRAM memory, running Ubuntu 16.04 LTS. Each core possesses an L1 data cache with 32KB memory and a 1024KB L2 cache. All cores share an 8448KB L3 cache.

**Dataset:** We use three real-world datasets for evaluation: 1) CAIDA [52]: IP traffic traces collected at Equinix-Chicago, specifically CAIDA15, CAIDA16, and CAIDA18 from 2015, 2016, and 2018, respectively, with 0.45M, 0.64M, and 1.29M items. 2) MAWI [53]: a dataset by the MAWI group analyzing Japanese wide area networks. We select a 15-minute 2022 trace with approximately 19.58M items. 3) Campus [54]: gathered from a campus DNS network with over 4000 users during peak hours for 10 days in April-May 2016. We randomly choose a trace from April with 0.87M distinct items.

**Benchmarks:** For detecting *heavy hitters* and *heavy changers*, we conduct a comparative evaluation of Stable-Sketch against nine existing approaches: (1) *probability-based* methods including CocoSketch [29], USS [30], RAP [34], and PRECISION [35]; (2) *non-probability-based* methods including MV-Sketch [7], Elastic [39], CMHeap [6], CountHeap [36], and Space-Saving [33]. For RAP and PRECISION, the number of arrays is configured as 2 [35]. For MV-Sketch, we set the number of rows to 4 [7, 56]. The parameters for the rest of the schemes are aligned with [29]. More details about these benchmarks are discussed in Section 6. We configure the default threshold $\theta$ as 0.0005, meaning that if the item frequency is over $\theta N$, it will be identified as a heavy hitter. The threshold of heavy changer detection is consistent with finding heavy hitters [22]. For *persistent item lookup*, we pick three baselines: Small-Space (SS) [19], WavingSketch [22], and On-Off Sketch [18]. Since PIE [20] only works under large memory allocations, we omit a comparison here. The number of key-value pairs in On-Off Sketch and the number of cells in WavingSketch are both set as 16 [22]. We divide each dataset into 1,600 time windows [18] and set the threshold $\phi$ to 0.5, indicating that if an item appears over 800 windows, it will be recognized as persistent. Notice that we also *alter the threshold for different detection tasks* to verify Stable-Sketch's robustness in Appendix C.3.

**Implementation:** We implement Stable-Sketch and other approaches in C++ and use the source-destination address pairs as item keys (64 bits). For all the traffic, we concentrate on the IPv4 items only and adopt MurmurHash [57] to hash these items into the sketch. We fix the number of rows $m = 4$ [7, 56] and adjust $u$ according to the pre-allocated memory size.

**Metrics:** 1) Precision: the ratio of correctly reported items to all reported ones; 2) Recall: the ratio of correctly reported items to all correct items; 3) F1 score: $\frac{2 \times recall \times precision}{recall+precision}$; 4) Average Relative Error (ARE): $\frac{1}{|\Omega|} \sum_{f \in \Omega} \frac{|S(f)-\hat{S}(f)|}{S(f)}$, where $\Omega$ is the set of true heavy/persistent items reported; 5) Update Throughput: the update speed of the scheme expressed in million operations (insertions) per second (Mops). We conduct each experiment five times and choose median values as in [29].

### 5.1 Detection Accuracy on Different Tasks

*5.1.1 Heavy Hitter Detection.* Figures 4–5 compare the detection performance of Stable-Sketch with that of the benchmarks

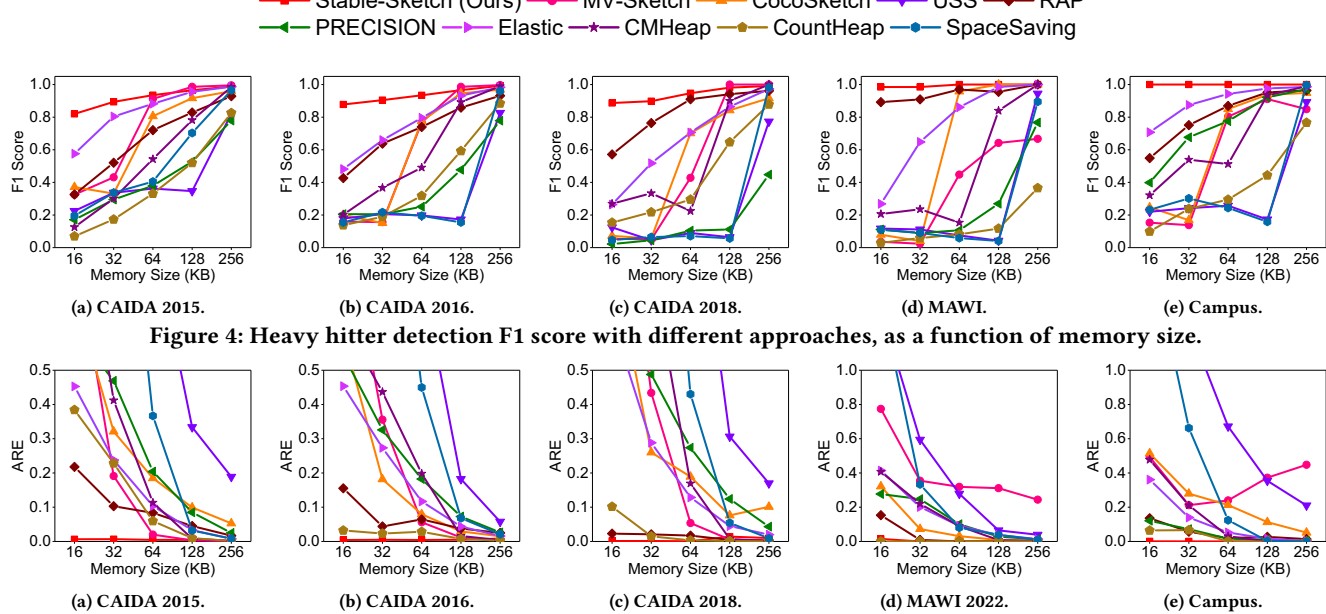

Figure 4: Heavy hitter detection F1 score with different approaches, as a function of memory size.

Figure 5: Heavy hitter detection ARE with different approaches, as a function of memory size.

considered, across three authentic datasets: CAIDA, MAWI, and Campus.

As illustrated in Figure 4, Stable-Sketch consistently achieves the highest F1 score across all settings, demonstrating an average improvement over existing algorithms ranging from 9.45% to 139.81% on the CAIDA 2015, 18.29% to 188.23% on the CAIDA 2016, 12.75% to 542.19% on the CAIDA 2018, 5.19% to 664.62% on the MAWI dataset, and 11.45% to 180.19% on the Campus dataset.

*The remarkable F1 score performance of Stable-Sketch can be attributed to its exceptional precision and commendable recall rates. Due to space constraints, we have relocated the results and analysis of recall and precision to Appendix C.1.* In summary, Stable-Sketch consistently maintains a precision score close to 1 across various memory budgets, outperforming existing approaches (as shown in Figure 14). This high precision is achieved through the use of multidimensional features (item and bucket statistics) and the probabilistic eviction of items stored in buckets. Stable-Sketch effectively prevents heavy hitters from being easily replaced by other items, even with limited available memory (16KB). Furthermore, we observe that for USS and SpaceSaving, precision decreases as memory size increases from 16KB to 128KB. This is due to their aggressive eviction of items stored in buckets, leading to more non-heavy items being incorrectly identified as heavy hitters with larger memory, resulting in reduced precision. RAP and PRECISION make replacement decisions based on probabilities computed from item frequency, which does not offer adequate protection for heavy items in highly skewed data streams, especially with tight L1 memory constraints, resulting in lower precision compared to Stable-Sketch. Figure 15 also demonstrates that Stable-Sketch maintains a commendable recall rate across different traces when compared to the baseline methods.

Additionally, Stable-Sketch demonstrates exceptionally low estimation error, with values close to zero in all memory settings (Figure 5). For example, when compared with RAP, Stable-Sketch

reduces the ARE by 1837.63% in CAIDA 2015, 1103.7% in CAIDA 2016, 147.21% in CAIDA 2018, 1001.16% in the MAWI, and 25636.04% in the Campus traces on average.

Furthermore, we extend our evaluation of Stable-Sketch to encompass additional datasets and conduct comparisons with more advanced approaches to assess its performance comprehensively. For a deeper dive into this extended evaluation, please refer to Appendix C.2.

*5.1.2  **Heavy Changer Detection**.* To illustrate the performance of heavy changer detection, we utilize the CAIDA 2018 trace as an example. In Figure 6(a), we observe that Stable-Sketch achieves significantly higher F1 scores compared to RAP and Elastic, with improvements of 24.83% and 80.57%, respectively. In terms of estimation error, Stable-Sketch outperforms other schemes, as evident from the lowest ARE values shown in Figure 6(b).

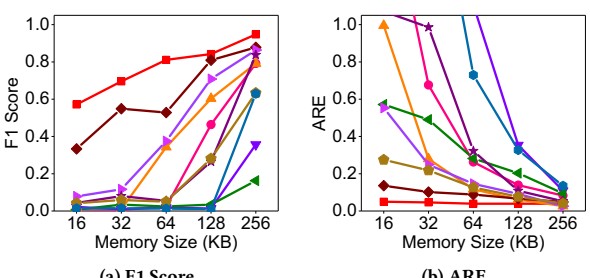

Figure 6: Heavy changer detection performance with different approaches (legend as in Figure 4).

*5.1.3  **Persistent Item Detection**.* Figure 7 provides insights into the F1 score and the ARE of persistent item lookup on the MAWI trace. It is evident that Stable-Sketch consistently maintains its optimality across different memory sizes. In comparison to the recent WavingSketch/On-Off Sketch approaches, Stable-Sketch achieves a remarkable increase in detection accuracy, with average improvements of 5428.75%/2852.76% on the MAWI trace. It is worth noting

that the performance of baselines considered is notably weaker on the MAWI dataset. This is primarily due to the heavier-tailed distribution in the MAWI dataset, which results in a smaller number of persistent items and increases the detection difficulty. Despite these challenges, Stable-Sketch consistently achieves the highest accuracy, thereby affirming its effectiveness in handling persistent item lookup tasks.

Besides, we also compare our method with the state-of-the-art On-Off Sketch with a memory size in the mega byte range. Specifically, we assess the performance of Stable-Sketch for persistent item lookup using the MAWI dataset, with memory sizes ranging from 1 to 1.3MB. The results reveal that Stable-Sketch achieves an F1 score between 0.981 and 0.991. In contrast, On-Off Sketch method yields F1 scores of 0.013 and 0.75 for the same memory sizes, which demonstrates the superior efficacy of Stable-Sketch.

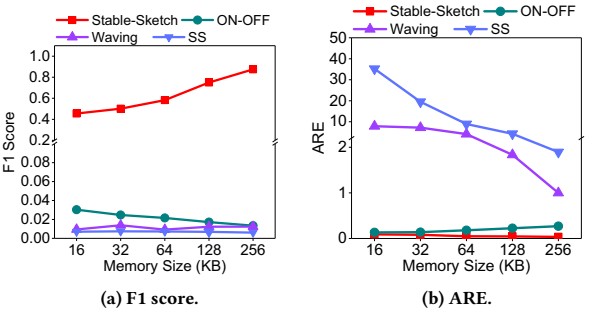

(a) F1 score.    (b) ARE.

**Figure 7: Persistent item detection performance with different approaches, as a function of memory size.**

## 5.2 Performance in Multiple Cases

*5.2.1 Accuracy under different thresholds.* To assess the robustness of Stable-Sketch, we vary threshold values for heavy item detection (0.0001 to 0.0021) and persistent item lookup (0.4 to 0.8). These experiments, conducted with CAIDA 2019 and new traces with varying skewness (0.2 and 0.8), consistently demonstrate our scheme's superior performance. Detailed detection accuracy analysis for various threshold settings is available in Appendix C.3.

*5.2.2 Ablation Study.* In our evaluation, we examine the effectiveness of each component of Stable-Sketch, including the replacement mechanism based on multi-dimensional features and the avoidance of redundant hash operations when an incoming item finds an available bucket. Additionally, we assess the impact of different eviction probability formulations on detection accuracy. For a comprehensive analysis of these components and details, please refer to Appendix C.4.

*5.2.3 Stable-Sketch with Fingerprint.* To assess the performance of Stable-Sketch with fingerprint, a detailed analysis is provided in Appendix C.5.

## 5.3 Processing Speed

*5.3.1 Update Speed.* We now evaluate the update speed of Stable-Sketch, taking heavy hitter and persistent item detection as examples. Figure 8 shows the update speed for different algorithms under the CAIDA 2018 trace. Results on other datasets exhibit similar trends. Observe in Figure 8(a) that Stable-Sketch's throughput surpasses that of all existing schemes for heavy hitter detection, with an improvement of 16.01% on average over MV-Sketch. Since

counter-based approaches, such as RAP and SpaceSaving, usually depend on pointers for finding the minimum item to replace, resulting in a lower update speed. As reported in Figure 8(b), the average update throughput of Stable-Sketch is 25.57% higher than that of the state-of-the-art method On-Off Sketch. This stems from two aspects: 1) Stable-Sketch leverages a compact data structure that does not lean on supplementary heaps or Bloom filters [23], which reduces the number of memory accesses; 2) Stable-Sketch abandons hash operations once an item finds an available bucket, mitigating the number of hash operations and guaranteeing a fast update speed.

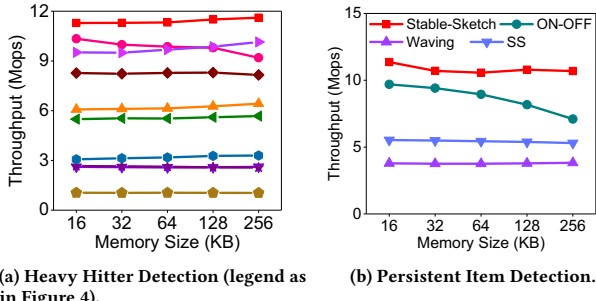

(a) Heavy Hitter Detection (legend as in Figure 4).    (b) Persistent Item Detection.

**Figure 8: Update speed with different approaches.**

*5.3.2 Query Time.* We also compare the query time of several advanced schemes returning all heavy items across different datasets. As shown in Figure 9, since Stable-Sketch is invertible and does not require excessive hash operations during the query process, its query time is smaller than that of existing schemes. In contrast, MV-Sketch requires additional hash operations for query, leading to a longer query time. Stable-Sketch also maintains its good performance when returning persistent items (results omitted due to the space limitation).

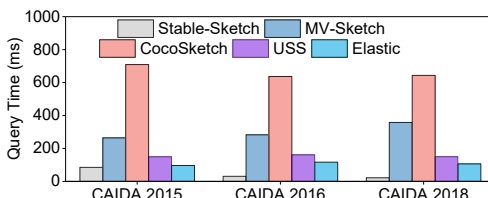

**Figure 9: Query time for heavy items (memory size: 32KB).**

## 5.4 Accelerating the Update Speed with SIMD Instructions

We further accelerate the update speed of Stable-Sketch with SIMD instructions [31], allowing us to process sequential operations in parallel. When a new item arrives, we utilize the primitive `MurmurHash3_x64_128` to calculate the hash value based on the item key and divide the hash value into $m$ parts. Afterward, unlike the vanilla Stable-Sketch inspecting each row individually to find an available bucket, we use the SIMD primitive `_mm256_cmpeq_epi64` to compare in parallel the newly arrived item's key with items recorded in $m$ rows. In this manner, Stable-Sketch with SIMD instructions only requires 1 step to find an available bucket for a newly arrived item, mitigating redundant comparison operations.

We compare the update speed with the CAIDA 2016 trace. Observe in Figure 10, where we find that with the aid of SIMD, Stable-Sketch significantly improves the update throughput on average by 78.84% and 46.55% over vanilla Stable-Sketch for the heavy hitter and persistent item detection, respectively, confirming the effectiveness of SIMD instructions. Additionally, it is important to note that as the memory budget expands, it eventually exceeds the capacity of the fastest cache level (L1), necessitating data retrieval from slower caches or main memory. This shift in memory access results in increased latency, reducing the data processing throughput.

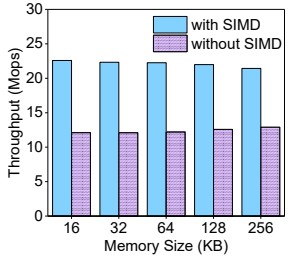 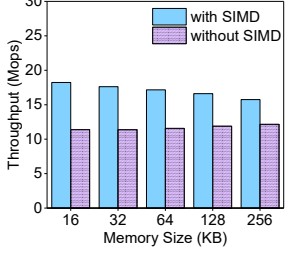

(a) Heavy Hitter Detection.           (b) Persistent item Detection.

**Figure 10: Update speed comparison w/wo SIMD instructions.**

## 5.5 Stable-Sketch Deployment in Practice

Here, we illustrate the practical feasibility of deploying Stable-Sketch on a programmable switch with minimal overhead. Our assessment of resource utilization reveals that Stable-Sketch conserves sufficient resources for other applications, affirming its viability for deployment on commercial hardware. For detailed implementation and evaluation results, please refer to Appendix C.6.

## 6 RELATED WORK

We briefly introduce existing schemes for different detection tasks and highlight their drawbacks, which inspired our design.

**Heavy Item Detection:** Existing approaches can be categorized into counter- and sketch- based [5]. Counter-based schemes aim to reduce memory usage by replacing the smallest recorded counter item with the newly arrived one. Space-Saving [33] employs multiple counters, updating the corresponding counter when a new item matches an existing one, or evicting the item with the smallest counter value. However, limited memory and hash collisions can lead to incorrect replacements. Unbiased Space Saving (USS) [30] builds upon Space-Saving by minimizing variance to achieve unbiased estimation, but it still struggles with lookup accuracy under memory constraints. Random Admission Policy (RAP) [34] enhances detection accuracy by probabilistically replacing counters with the smallest value. PRECISION [35] employs partial recirculation, either probabilistic or deterministic, for a fraction of packets from unmonitored streams. These schemes are non-invertible, necessitating a full item key space scan to recover heavy items, resulting in high memory access overhead. Additionally, most counter-based methods use pointers for finding the minimal element during updates, leading to low update throughputs.

Unlike counter-based methods, sketch-based approaches hash items into memory entries, summarizing cumulative information for efficient updates and low memory utilization at the expense of bounded errors. Count-min Sketch [6] hashes items into buckets,

estimating size based on the minimum bucket value, while Count Sketch [36] uses the average bucket value for estimation. Count Sketch Heap extends Count Sketch with a heap to track heavy candidates and their estimated values. However, under small memory sizes, hash collisions can lead to overestimating non-heavy items, reducing lookup precision. These methods are also non-invertible, resulting in slower query speeds. MV-Sketch [7] employs majority voting for invertible heavy item tracking. A-Sketch [10] introduces dynamic pre-filtering to identify and aggregate heavy items. Heavykeeper [8] balances space and accuracy using count-with-exponential-decay, actively evicting small items while preserving large ones. Cold Filter [41] distinguishes cold and hot items, using a separate structure for hot item frequencies. Loglog Filter [42] utilizes register arrays to filter cold items, approximating their sum of frequencies. HeavyGuardian [21] isolates hot items, maintaining large counters for them and small counters for cold items. Elastic Sketch [39] consists of heavy and light parts to manage heavy and non-heavy items separately. CocoSketch [29] leverages "power-of-$d$ choices" [44, 45] and probabilistically replaces items stored in buckets. However, when making replacement decisions only based on item information, heavy items are easily replaced by non-heavy ones with a limited memory.

**Persistent Item Detection:** Recent schemes can be divided into three categories: sample-, coding-, and sketch-based. Sample-based approaches, like Small-Space (SS) [19], configure a hash filter to record the occurrence of items based on a sampling rate. However, the sampling rate needs to be low to support small memory usage, amplifying detection errors. Even if sample-based methods try to track only potentially persistent items, they may still record many non-persistent ones, which take up a large portion of the available memory. Coding-based schemes, such as the Persistent items Identification schemE (PIE) [20], utilize a compact hash-based structure and Raptor codes to improve memory usage. However, PIE requires encoding and storing all items, regardless of potential persistence. For enhanced detection accuracy and memory efficiency, On-Off sketch [18] employs a compact data structure with a state field for each counter, periodically increasing an item's persistence. Nevertheless, this approach may misclassify many non-persistent items as persistent due to its coarse isolation method. WavingSketch [22] aims for unbiased estimation and uses a Bloom filter [23] for persistent item detection but suffers from severe false positives in cases of limited memory, leading to reduced lookup accuracy.

## 7 CONCLUSIONS

In this paper, we introduced Stable-Sketch, a versatile and effective sketch for item lookup, which maintains a fast processing speed and reaches high detection accuracy even with tight memory budgets (L1 cache). Specifically, Stable-Sketch utilizes a probability-based approach to discard items stored in buckets, considering both item and bucket statistics. We conducted extensive experiments on diverse datasets to evaluate the performance of Stable-Sketch with different detection tasks. The experimental results demonstrate that Stable-Sketch outperforms competing schemes, exhibiting superior processing speed and significantly improving the detection accuracy across various detection tasks. Moreover, we illustrated how to speed up our solution with SIMD instructions. Lastly, we demonstrated that it is feasible to deploy our solution in practice.

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

# A  DEFINITION OF SYMBOLS

### Table 2: Summary of frequently used symbols.

| Symbol | Definition |
|--------|------------|
| $f$ | an item in data stream $Q$ |
| $N$ | the total frequency of all items in data stream $Q$ |
| $m$ | the number of rows in the sketch |
| $u$ | the number of buckets in each row |
| $V(f)$ | frequency of the item $f$ |
| $P(f)$ | persistence of the item $f$ |
| $G$ | the number of time windows |
| $L$ | the eviction decay probability |
| $\theta$ | heavy hitter detection parameter, ranges in (0,1) |
| $\psi$ | heavy changer detection parameter, ranges in (0,1) |
| $\phi$ | persistent item detection parameter, ranges in (0,1) |

# B  MATHEMATICAL ANALYSIS

## B.1  No Over-estimation Error

THEOREM B.1. *For any given item $f$, let $V_t(f)$ and $\hat{V}_t(f)$ denote the actual frequency and estimated frequency at a particular time $t$, respectively. Then $\hat{V}_t(f) \leq V_t(f)$.*

PROOF. At the start of the detection task ($t = 0$), both $\hat{V}_t(f)$ and $V_t(f)$ are zero, so the theorem holds. Assume that at time $t - 1$, $\hat{V}_{t-1}(f) \leq V_{t-1}(f)$. At time $t$, two scenarios are possible: (i) if the incoming item is $f$ again, then $\hat{V}_t(f) = \hat{V}_{t-1}(f) + 1$ and $V_t(f) = V_{t-1}(f) + 1$. Hence, $\hat{V}_t(f) \leq V_t(f)$ is true; (ii) if an item other than $f$ arrives, the estimated frequency of item $f$ either decreases by 1 or stays the same, i.e., $\hat{V}_t(f) = \hat{V}_{t-1}(f) - 1$ or $\hat{V}_t(f) = \hat{V}_{t-1}(f)$. Given that $V_t(f) = V_{t-1}(f)$, it follows that $\hat{V}_t(f) \leq V_t(f)$. Since the claim holds for all scenarios, Theorem B.2 is proven.  □

## B.2  Error Bound of Stable-Sketch

To derive the error bound, we make an assumption that is generally valid (confirmed in Appendix B.3.2): once a heavy item enters a bucket, it remains in the bucket until the detection task is complete. Then we get the error bound of Stable-Sketch as

THEOREM B.2. *Given a small positive number $\beta$ and a heavy item $f$ with frequency $V(f)$, the inequality $\Pr \left\{ V(f) - \hat{V}(f) \geq \lceil \beta N \rceil \right\} \leq \frac{[\ln(V(f)) + \varphi]}{\beta N \ln(S)}$ holds, where $\varphi$ denotes the Euler-Mascheroni constant, $S$ denotes the bucket stability that records item $f$, and $N$ represents the total number of entries for all items.*

PROOF. When an item distinct from $f$ arrives and maps to the same bucket $B(i, j)$ as $f$, the value counter of this bucket undergoes either a decrement of 1 or remains unaltered. We use $G_{i,j}$ to denote the times in which items distinct from $f$ hash into the same bucket. Consequently, we infer that $V(f) - G_{i,j} \leq \hat{V}(f) \leq V(f)$, where $\hat{V}(f)$ is equivalent to $B(i, j).V$. We employ a random variable $D_{i,j,x}$ to represent the event where the value counter of bucket $B(i, j)$ decreases by 1 upon the arrival of the $x$-th item, where $1 \leq x \leq G_{i,j}$. Hence, $\hat{V}(f) = V(f) - \sum_{x=1}^{G_{i,j}} D_{i,j,x}$. By applying the Markov inequality in conjunction with a small positive value $\beta$, we deduce:

$$\Pr \left\{ \hat{V}(f) \leq V(f) - \beta N \right\} = \Pr \left\{ V(f) - \sum_{x=1}^{G_{i,j}} D_{i,j,x} \leq V(f) - \beta N \right\}$$

$$= \Pr \left\{ \sum_{x=1}^{G_{i,j}} D_{i,j,x} \geq \beta N \right\} \leq \frac{\mathbb{E} \left[ \sum_{x=1}^{G_{i,j}} D_{i,j,x} \right]}{\beta N}.$$

Assuming that the distribution of packets from all items is uniform, we can derive the following:

$$\mathbb{E} \left[ \sum_{x=1}^{G_{i,j}} D_{i,j,x} \right] = \mathbb{E} \left[ D_{i,j,x} G_{i,j} \right] = \sum_{G_{i,j}}^{V(f)} p(G_{i,j}) \left[ G_{i,j} \mathbb{E}(D_{i,j,x} | G_{i,j}) \right].$$

Let $\omega$ denote the final value of the counter in bucket $B(i, j)$ when the detection task is complete. Under the assumption that the arrival probability of an item is constant, ranging from 1 to $\omega$, we obtain

$$\mathbb{E}(D_{i,j,x} | \omega) = \sum_{V=1}^{\omega} \frac{1}{\omega} \frac{1}{(V \times S) + 1} < \sum_{V=1}^{\omega} \frac{1}{\omega} \frac{1}{V \times S},$$

where $V$ and $S$ represent the value counter and bucket stability counter of bucket $B(i, j)$, respectively.

Let $p(\omega)$ denote the probability that $\omega$ is any of the values in the $[V(f) - G_{i,j}, V(f)]$ range, then

$$\mathbb{E}(D_{i,j,x} | G_{i,j}) < \sum_{\psi = V(f) - G_{i,j}}^{V(f)-1} p(\omega) \sum_{V=1}^{\omega} \frac{1}{\omega} \frac{1}{V \times S}$$

$$\leq \sum_{\psi = V(f) - G_{i,j}}^{V(f)-1} p(\omega) \sum_{V=1}^{\omega} \frac{1}{V(f) - G_{i,j}} \frac{1}{V \times S}$$

$$\leq \sum_{\psi = V(f) - G_{i,j}}^{V(f)-1} p(\omega) \sum_{V=1}^{V(f)} \frac{1}{V(f) - G_{i,j}} \frac{1}{V \times S} = \frac{1}{V(f) - G_{i,j}} \sum_{V=1}^{V(f)} \frac{1}{V \times S}.$$

Then we get

$$\mathbb{E} \left[ \sum_{x=1}^{G_{i,j}} D_{i,j,x} \right] = \mathbb{E} \left[ D_{i,j,x} G_{i,j} \right] = \sum_{G_{i,j}}^{V(f)} p(G_{i,j}) \left[ G_{i,j} \mathbb{E}(D_{i,j,x} | G_{i,j}) \right]$$

$$\leq \sum_{G_{i,j}=1}^{V(f)-1} p(G_{i,j}) \left( \frac{G_{i,j}}{V(f) - G_{i,j}} \sum_{V=1}^{V(f)} \frac{1}{V \times S} \right).$$

Since item $f$ is a heavy item with a large value, the distribution of $G_{i,j}$ can be approximated as a Poisson distribution with a mean of $\frac{N}{h} p(G_{i,j}) = \frac{N}{h} e^{-\frac{N}{h} G_{i,j}}$, where $h$ denotes the number of buckets in each row. Consequently, we can derive the following:

$$\mathbb{E} \left[ \sum_{x=1}^{G_{i,j}} D_{i,j,x} \right] \leq \sum_{G_{i,j}=1}^{V(f)-1} \frac{N}{h} e^{-\frac{N}{h} G_{i,j}} \left( \frac{G_{i,j}}{V(f) - G_{i,j}} \sum_{V=1}^{V(f)} \frac{1}{V \times S} \right)$$

$$= \sum_{V=1}^{V(f)} \frac{1}{V \times S} \left[ \sum_{G_{i,j}=1}^{\frac{V(f)}{2}} \frac{N}{h} e^{-\frac{N}{h} G_{i,j}} \left( \frac{G_{i,j}}{V(f) - G_{i,j}} \right) \right.$$

$$\left. + \sum_{G_{i,j}=\frac{V(f)}{2}+1}^{V(f)-1} \frac{N}{h} e^{-\frac{N}{h} G_{i,j}} \left( \frac{G_{i,j}}{V(f) - G_{i,j}} \right) \right]$$

$$\leq \sum_{V=1}^{V(f)} \frac{1}{V \times S} \left[ \sum_{G_{i,j}=1}^{\frac{V(f)}{2}} \frac{N}{h} e^{-\frac{N}{h} G_{i,j}} + \sum_{G_{i,j}=\frac{V(f)}{2}+1}^{V(f)-1} \frac{N}{h} e^{-\frac{N}{h} \frac{V(f)}{2}} \frac{G_{i,j}}{V(f) - G_{i,j}} \right]$$

$$\leq \sum_{V=1}^{V(f)} \frac{1}{V \times S} \left[ 1 + \sum_{G_{i,j}=\frac{V(f)}{2}+1}^{V(f)-1} \frac{N}{h} e^{-\frac{N}{h} \frac{V(f)}{2}} \frac{G_{i,j}}{V(f) - G_{i,j}} \right]$$

$$\leq \sum_{V=1}^{V(f)} \frac{1}{V \times S} \left[ 1 + \sum_{G_{i,j}=\frac{V(f)}{2}+1}^{V(f)-1} \frac{N}{h} e^{-\frac{N}{h} \frac{V(f)}{2}} \frac{V(f) - 1}{V(f) - (V(f) - 1)} \right]$$

$$\leq \sum_{V=1}^{V(f)} \frac{1}{V \times S} \left[ 1 + \sum_{G_{i,j}=\frac{V(f)}{2}+1}^{V(f)-1} \frac{N}{h} e^{-\frac{N}{h} \frac{V(f)}{2}} V(f) \right]$$

$$\leq \sum_{V=1}^{V(f)} \frac{1}{V \times S} \left[ 1 + V(f) \frac{N}{h} \frac{V(f)}{2} e^{-\frac{N}{h} \frac{V(f)}{2}} \right].$$

Since $V(f)$ is a large value, the term $\frac{N}{h} \frac{V(f)}{2} e^{-\frac{N}{h} \frac{V(f)}{2}}$ tends to 0, and $\sum_{V=1}^{V(f)} \frac{1}{V}$ can be approximated as $\ln(V(f)) + \varphi$, where $\varphi$ represents the Euler-Mascheroni constant, approximately 0.577. Hence, we can derive the following:

$$\mathbb{E}\left[ \sum_{x=1}^{G_{i,j}} D_{i,j,x} \right] \leq \sum_{V=1}^{V(f)} \frac{1}{V \times S} < \frac{1}{\ln(S)} \sum_{V=1}^{V(f)} \frac{1}{V} \approx \frac{1}{ln(S)} \left[ \ln(V(f)) + \varphi \right].$$

Finally, we get the estimation error bound as

$$\Pr\left\{ V(f) - \hat{V}(f) \geq \lceil \beta N \rceil \right\} \leq \Pr\left\{ \hat{V}(f) \leq V(f) - \beta N \right\}$$

$$\leq \frac{\mathbb{E}\left[ \sum_{x=1}^{G_{i,j}} D_{i,j,x} \right]}{\beta N} \leq \frac{\left[ \ln(V(f)) + \varphi \right]}{\beta N \ln(S)}.$$

$\square$

## B.3 Empirical Validation

*B.3.1 Correctness of Theorem 1.* To validate the accuracy of Theorem 1, we employ two traces obtained from the CAIDA 2015 and CAIDA 2018 datasets, comprising 15.85M and 14.96M packets, respectively. The experimental configuration is consistent with the setup in Section 5. Figure 11 presents the real and estimated frequency of heavy items with a memory size of 64KB. It can be observed that there are no overestimation errors, and the estimated frequency closely aligns with the real frequency, thus confirming the high estimation accuracy of Stable-Sketch.

*B.3.2 Reasonableness of the Assumption.* When deriving the error bound, we assume that a heavy item will remain in the bucket once it has entered. However, there are two scenarios where a heavy item may not always stay in the bucket: 1) it may be replaced by non-heavy items, and 2) it may be replaced by other heavy items. To analyze the impact of these scenarios, Figure 12 presents the frequency of occurrence where other items mistakenly evict heavy items during the update process. From the results, we observe that for Stable-Sketch, most incorrect replacement events are caused by non-heavy items. However, the number of wrong replacements is significantly smaller than with existing methods such as MV-Sketch

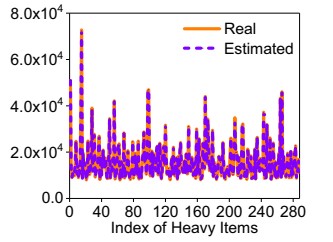 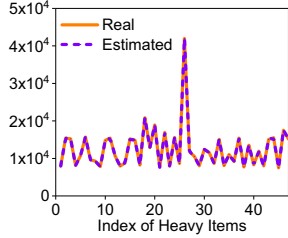

(a) Frequency (CAIDA 2015).  (b) Frequency (CAIDA 2018).

**Figure 11: Comparison between the estimated and real frequent of heavy items under different traces.**

(as shown in Table 1), indicating that Stable-Sketch effectively mitigates the risk of erroneously evicting heavy items from a bucket. Furthermore, we also consider the total number of replacement events during the update process. For example, for the CAIDA 2015 trace with a memory size of 16KB, the total number of replacement events is 340,362. In comparison, the number of wrong replacement events involving heavy items is only a tiny fraction of the overall replacements. This demonstrates that Stable-Sketch has a significantly lower probability of erroneously evicting heavy items from the bucket. Therefore, the assumption of heavy items remaining in a bucket holds in most cases.

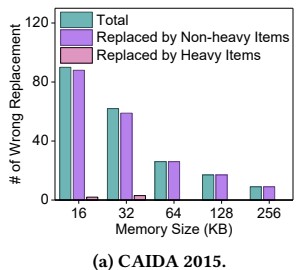 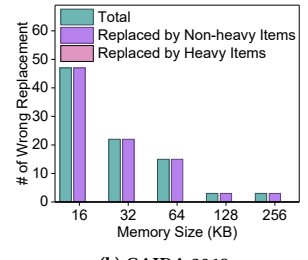

(a) CAIDA 2015.  (b) CAIDA 2018.

**Figure 12: Number of heavy items being wrongly evicted from a bucket during the update process under different traces.**

In these two exceptional cases, it becomes apparent that additional memory would be required to mitigate hash collisions, otherwise deriving an error bound becomes non-trivial. Fortunately, there are several techniques available to alleviate these limitations. One such technique involves utilizing the hash chain approach [38]. When a heavy item is evicted from the bucket by other items, we can attempt to relocate it by employing multiple hash operations to find an available bucket instead of discarding it directly. This increases the likelihood of heavy items being recorded in the bucket. However, it is important to note that this method necessitates additional hash operations, thus reducing the update speed. If update speed is not a critical concern for the user, such techniques can be additionally employed to further ameliorate the detection accuracy.

*B.3.3 Correctness of Theorem 2.* Based on the derived error bound, we observe that for a heavy item, the estimation error decreases as its bucket stability increases. *This finding highlights the importance of considering the bucket stability to minimize the estimation error.*

To validate the accuracy of the derived error bound, we conduct experiments using a CAIDA 2018 trace. We set $\beta$ to $2^{-21}$ and vary the memory size from 16KB to 256KB. Note that $\beta$ is a user-defined parameter, and here the $2^{-21}$ setting corresponds to a value of 8 for $\lceil \beta N \rceil$. This means that we calculate the probability of the error between the actual value and the estimated value of the heavy items greater than 8. The results shown in Figure 13 demonstrate that the empirical values are consistently smaller than the corresponding theoretical ones, which validates the accuracy of our theoretical analysis. In prac-

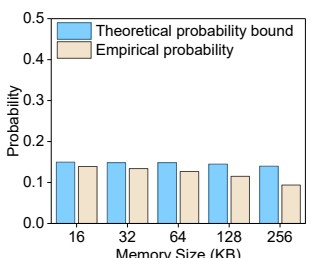

**Figure 13: Theoretical bound and empirical probability of Stable-Sketch.**

tice, different values can be chosen for $\beta$. We also configure $\beta$ as $2^{-20}$ and find that the results are consistent.

*B.3.4 Comparison with Existing Work.* There are alternative methods for heavy item detection, a classic one being the Count-Min Sketch. However, Count-Min Sketch is known to have overestimation errors, whereas the overestimation error of Stable-Sketch is 0. Moreover, Count-Min Sketch records the frequency of all items, whereas our Stable-Sketch focuses on tracking the frequency of potential heavy items. We also compare Stable-Sketch with an advanced probability-based method with underestimation errors, namely HeavyKeeper [8]. In the *worst-case* scenario, where the memory size is limited (e.g., 16KB), we observe that the error bound of Stable-Sketch is significantly smaller than that of HeavyKeeper when $\beta$ is set as $2^{-21}$ under the CAIDA 2018 trace. To further investigate the robustness of Stable-Sketch, we adjust the value of $\beta$ to $2^{-20}$ and find that the error bound for our method is 0.074, while that of HeavyKeeper is 0.572 when the memory size is 16KB. These observations validate the effectiveness of Stable-Sketch.

## C  EVALUATION

### C.1  Precision and Recall for Heavy Item Detection

As shown in Figure 14, the precision of Stable-Sketch is always around 1 under different memory budgets, which is much higher than that of existing approaches. The reason for this is that with the help of multidimensional features (item and bucket statistics) and the probabilistic eviction of items saved in buckets, Stable-Sketch can effectively prevent heavy hitters from being effortlessly replaced by other items, even when the available memory is small (16KB). We also observe that for USS and SpaceSaving, the precision degrades as the memory size increases from 16KB to 128KB. This is because they aggressively evict items stored in buckets, and the increasing memory size causes more non-heavy items to be incorrectly identified as heavy ones, resulting in reduced precision. RAP and PRECISION make replacement decisions based on probabilities computed by item frequency, which does not provide adequate protection for heavy items in highly skewed data streams, especially under tight L1 memories, resulting in lower precision than that

of Stable-Sketch. As observed from Figure 15, Stable-Sketch also maintains commendable recall rate on different traces.

### C.2  Stable-Sketch Deep Dive

1) We also evaluate the performance of Stable-Sketch using a synthetic intrusion dataset [55]. This dataset is specifically designed for intrusion detection evaluation and contains 4.16 million items, simulating various user behaviors. The experimental results demonstrate that Stable-Sketch consistently outperforms the considered baselines, achieving an improvement of 11.48% - 180.19% in identifying heavy items (figure omitted due to the space limitation).

2) In addition, we evaluate Stable-Sketch's detection accuracy against several advanced sketch-based methods for identifying heavy items, such as LadderFilter (Ladder) [40], Cold Filter (Cold) [41], Loglog Filter (Loglog) [42], A-Sketch [10], UA-Sketch [43], HeavyGuardian [21], DHS [11], and SALSA [12]. Ladder Filter [40] is a state-of-the-art scheme that discards approximately infrequent items using multiple LRU queues. Accordingly, we use LadderFilter, Cold Filter, and Loglog Filter with SpaceSaving, as described in [40]. UA-Sketch [43] utilizes the uninterrupted arrival counter to probabilistically evict items. SALSA [12] employs dynamic counters that use small counters initially and merges adjacent counters when they overflow. DHS [11] employs many fixed-size buckets. When an item in a smaller counter overflows, DHS will reallocate the space in the bucket and move the item to the larger counter to accommodate its increased frequency.

As illustrated in Figure 16, our scheme maintains high accuracy even with a limited memory budget of 16KB. When compared to the state-of-the-art LadderFilter, Stable-Sketch achieves an average improvement of 78.64% in detection accuracy, owing to its efficient replacement strategy based on multi-dimensional features. Additionally, Stable-Sketch outperforms DHS and SALSA, which dynamically resize their counter sizes, with an average improvement in F1 score of 41.6% and 81.47%, respectively. This highlights the effectiveness of Stable-Sketch, even with fixed-size counters, given that the number of heavy items is typically very small in practice.

### C.3  Detection with Different Thresholds

To assess the robustness of Stable-Sketch, we set the memory size to 32KB and vary the threshold from 0.0001 to 0.0021 for heavy item detection using a larger public trace (CAIDA 2019) and traces with different levels of skewness (0.2 and 0.8). The CAIDA 2019 trace comprises 1.52M items, while the traces with skewness 0.2 and 0.8 consist of 7.53M and 7.34M items, respectively.

As illustrated in Figures 17(a)-(c), Stable-Sketch consistently outperforms competitive approaches, MV-Sketch and CocoSketch, across various threshold settings. This validates the effectiveness and robustness of Stable-Sketch. In addition, we examine the effects of varying the threshold for persistent item detection (from 0.4 to 0.8). The results presented in Figure 17(d) demonstrate that Stable-Sketch maintains its superiority over the most competitive approach, On-Off Sketch. Furthermore, we observe a decreasing trend in the accuracy of On-Off Sketch as the threshold value increases. This can be attributed to the fact that, as the threshold increases, the number of persistent items decreases. Under tight

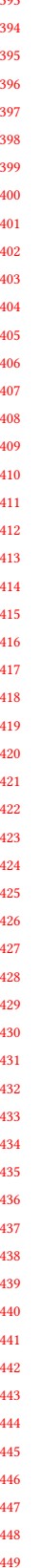

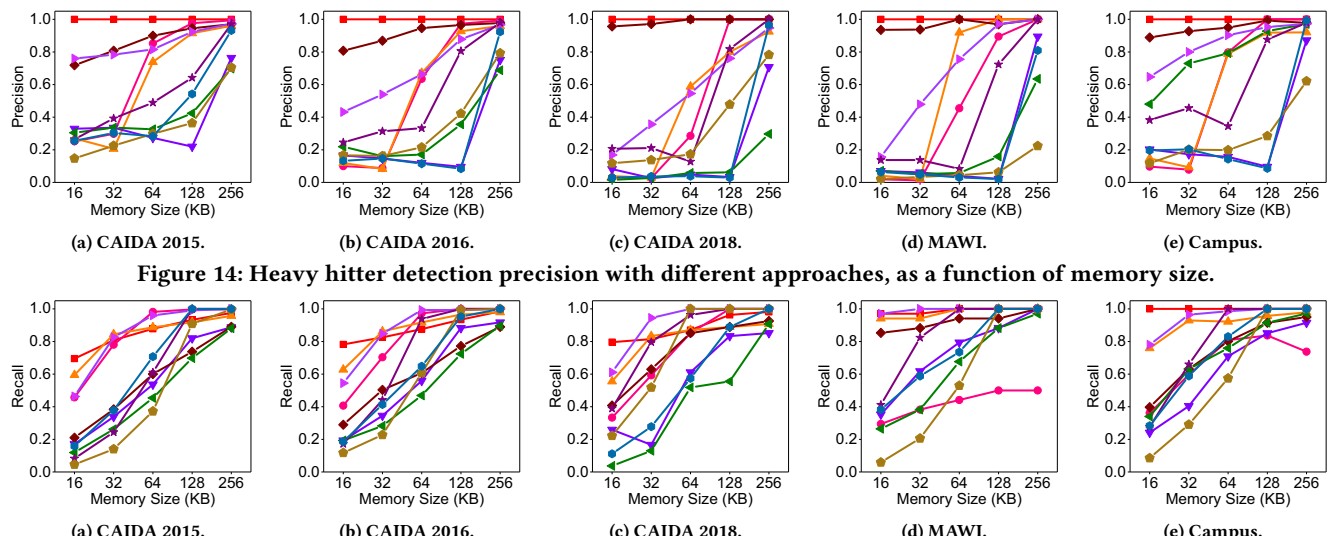

Figure 14: Heavy hitter detection precision with different approaches, as a function of memory size.

Figure 15: Heavy hitter detection recall with different approaches, as a function of memory size.

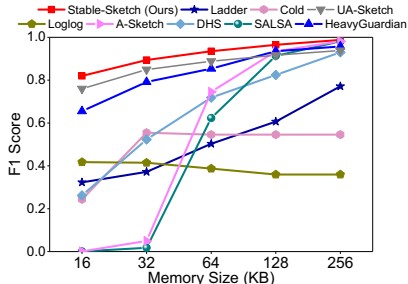

Figure 16: Heavy hitter detection F1 score with advanced approaches, as a function of memory size (CAIDA 2015).

memory allocation and excessive hash collisions, the naive replacement strategy employed by On-Off Sketch results in numerous persistent items being erroneously replaced by non-persistent ones, thus leading to a decline in its detection performance.

## C.4 Deep Diving into Stable-Sketch's Operation

Stable-Sketch builds on three core design insights: replacing stored items using multi-dimensional information, stopping hash operations on time and evicting items tracked in buckets based on a probability $L(f)$. Here, we take persistent item lookup as an example and utilize the CAIDA traces to investigate the contribution of each principle to Stable-Sketch's performance.

*C.4.1 Multi-dimensional Information.* We set the memory size to 16KB. Figure 18(a) validates the importance of considering bucket stability. Compared with a Stable-Sketch version focusing on single-dimensional information only (persistence value), Stable-Sketch with stability can deliver more protection to persistent items from being expelled by non-persistent ones under reduced memory sizes, with a reduction of estimation error by 72.82% on average. Though recording bucket stability increases the storage overhead, as shown in Figure 18(b), the update throughput only experiences a slight

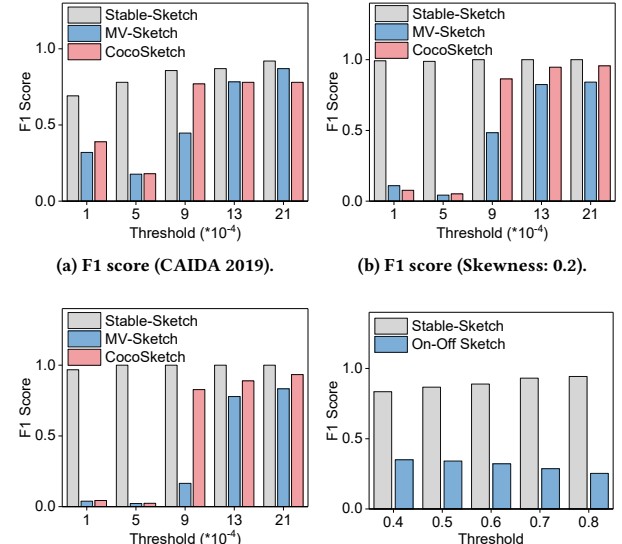

(a) F1 score (CAIDA 2019).  (b) F1 score (Skewness: 0.2).

(c) F1 score (Skewness: 0.8).  (d) F1 score (CAIDA 2019, Persistent).

Figure 17: Detection performance under different thresholds.

decrease, meaning that its advantage of guarding persistent items out weights the overhead.

*C.4.2 Abandoning Redundant Hash Operations.* Compared with sketches that map each item to all rows, abandoning hash operations on time can save memory space and thus allows storing more items. As shown in Figure 19, this leads to a 7.35% increase in detection accuracy. Moreover, eliminating redundant hash operations reduces update time, leading to a 17.62% improvement in update throughput.

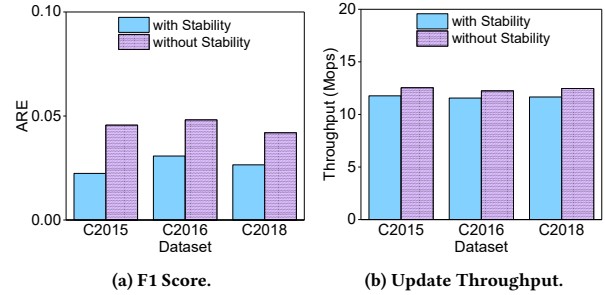

**(a) F1 Score.**    **(b) Update Throughput.**

**Figure 18: Estimation accuracy and update speed comparison w/wo considering bucket stability (C stands for CAIDA).**

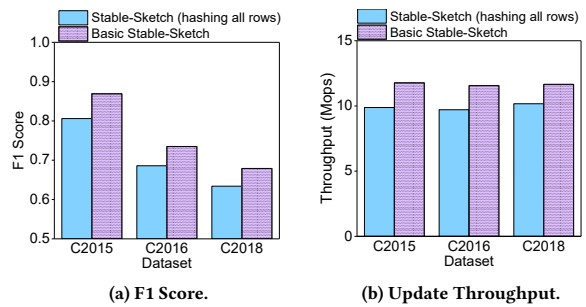

**(a) F1 Score.**    **(b) Update Throughput.**

**Figure 19: Accuracy and update speed comparison with different hash methods (C stands for CAIDA).**

*C.4.3    Different Replacement Probabilities.* Stable-Sketch leverages a default decay probability of $\frac{1}{B(R,M).V \times B(R,M).S+1}$ to evict existing items recorded in buckets. Here, we evaluate the impact of different replacement methods on Stable-Sketch's detection performance using the CAIDA 2015 and 2016 traces. Specifically, we examine three forms: 1) Additive denominator (Add) replacement, which replaces the recorded item directly with a probability of $\frac{1}{B(R,M).V+B(R,M).S+1}$; 2) Expo_Multi, which decays the value counter is decreased based on the probability $\kappa^{B(R,M).V \times B(R,M).S}$ (where $\kappa$ is a constant set to 1.08 [8]); when the value counter reaches 0, the new incoming item replaces the incumbent item tracked in the bucket; and 3) Expo_Add, which decays the value counter with a probability of $\kappa^{B(R,M).V+B(R,M).S}$. As listed in Figure 20, we find that using our default replacement probability provides the highest F1 score for Stable-Sketch compared to the other three replacement methods. On average, this default approach exhibits an improvement of 19.92%, 27.5% and 34.31% in terms of F1 score over the Add, Expo_Multi and Expo_Add methods over the CAIDA 2015 trace, respectively.

## C.5    Stable-Sketch with Fingerprint

Stable-Sketch tracks an item's key in each bucket, but a longer key (such as 5-tuples instead of source-destination pairs in network task scenarios) can consume valuable memory resources. To optimize memory usage, we propose a variant called Stable-Sketch* that only tracks the item's fingerprint instead of the entire key. Fingerprint $h_g(f)$ of an item $f$ is a hash value produced by a specific hash function $h_g$. Although it is possible for hash collisions to occur among items, the likelihood of such events is relatively low and

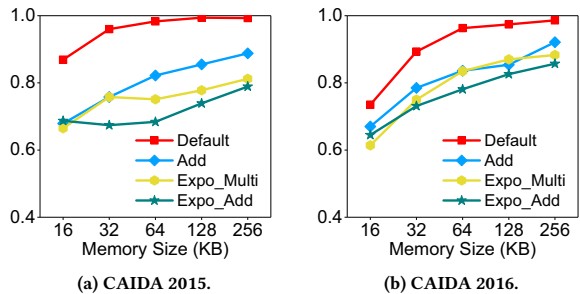

**(a) CAIDA 2015.**    **(b) CAIDA 2016.**

**Figure 20: F1 score with different replacement strategies.**

can be neglected. If the fingerprint size is set to 32 bits and there are 340 buckets in each row, for a dataset with 1,000,000 items, the probability of fingerprint collisions is $6.85 \times 10^{-7}$, which is considerably low [21].

Figure 21 demonstrates the detection accuracy and update speed for heavy item lookup using the CAIDA 2015 trace. The results in Figure 21(a) indicate that using fingerprints improves the recall by 4.83% under tight memory settings (16KB). This is due to the more efficient usage of memory resources, which allows Stable-Sketch to track more items. Precision is not shown since it remains high regardless of whether fingerprints are used or not. Overall, the F1 score is improved by 2.8% with a 16KB memory allocation. However, using fingerprints slows down the update throughput due to the additional hash operation required to obtain the fingerprint value of each item. As shown in Figure 21(b), compared to vanilla Stable-Sketch, Stable-Sketch* sees an average update throughput drop of 3.77%. The query time also increases, since it needs extra hash operations to retrieve items. This fingerprint-based Stable-Sketch* variant provides users with an alternative option, if detection accuracy is to be prioritized. Otherwise, opting for the default Stable-Sketch enables striking a good balance between detection accuracy and update throughput.

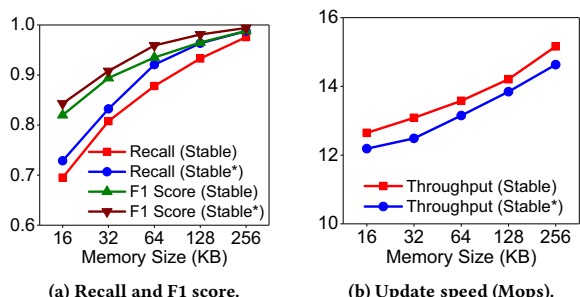

**(a) Recall and F1 score.**    **(b) Update speed (Mops).**

**Figure 21: Detection accuracy and update speed comparison with/without fingerprint, as a function of memory size.**

## C.6    Stable-Sketch Deployment in Practice

In this subsection, we demonstrate that it is feasible to deploy Stable-Sketch in practice with limited overhead. Specifically, we implement Stable-Sketch into a programmable switch with P4 [59], a high-level language for programming protocol-independent packet

**Table 3: Resource usage of Stable-Sketch.**

| Resource | Usage | Percentage |
|---|---|---|
| Match Crossbars | 96 | 6.81% |
| Hash Bits | 156 | 3.41% |
| SRAM (KiB) | 27 | 3.07% |
| ALUs | 9 | 20.46% |
| Gateways | 23 | 13.06% |
| VLIW Instructions | 25 | 7.1% |

processors, and compile it with P4 Studio [60]. Compared to other hardware such as Field Programmable Gate Arrays (FPGAs) [61], programmable switches are renowned for their high processing speeds and strict design constraints. We construct each row as an array of registers. For each row, we use different hash functions to map items, like *crc_16*, *crc_16_dect*, and *crc_16_dds_110*. For an item that fails to find an available bucket, we resubmit this item once using the *recirculation* primitive, as the register can only be accessed once during each update process [29, 56]. Then, we test whether this newly arrived item can replace the recorded item based on the value counter and bucket stability. Considering the memory access limitation of the programmable switch, we track

the item key and the value of counters into temporary metadata when a hash collision occurs. Then we can find the bucket with the smallest frequency value by comparing these values individually. If the replacement is successful, the key field in the corresponding register will be rewritten with the key of the newly arrived item; otherwise, the switch will discard the new arrival item directly.

Table 3 lists the switch resource usage of Stable-Sketch for heavy hitter detection. The operation of Stable-Sketch is achieved by the match-action pipeline, which requires the crossbar to extract match keys and action inputs from the item header vector and thus consumes 6.81% of match crossbar resources. Due to the hash operation for each item, Stable-Sketch accounts for 3.41% of the hash bits. Each pipeline stage owns SRAM that can be used to maintain state, like counter arrays [58]. Stable-Sketch occupies 3.07% of the total SRAM resources. In addition, the ALU (arithmetic logic unit) can be employed for counter update operations such as counter increment. Stable-Sketch uses 9 ALUs, which takes up 20.46% of the total ALUs. For other types of resources, the maximum demand of Stable-Sketch is no more than 14% of the entire budget. These results confirm that Stable-Sketch leaves adequate resources to be used for other applications, indicating that it is feasible to deploy our solution on commercial hardware, such as programmable switches.

