# OpenReview forum: "Stable-Sketch: A Versatile Sketch for Accurate, Fast, Web-Scale Data Stream Processing"
_ACM.org/TheWebConf/2024/Conference — TheWebConf24 Oral_

### Official Review · Reviewer_8WbR · 2023-11-22

**Novelty:** 5
**Technical Quality:** 5

**Review:**

Summary of the review:

The paper studies three important problems in the domain of web-scale data stream processing: heavy hitter detection (finding items with frequency greater than a threshold), heavy changer detection  (finding items with drastically different frequencies in two consecutive time windows), and persistent item detection (finding items that are persistent with respect to a given threshold). The focus of the paper is on sketch based algorithms.

The main contributions of the work can be summarized as follows.

1. The authors propose a novel algorithm called Stable-Sketch that can accurately and efficiently detect heavy hitters, heavy changers, and persistent items in data streams.

2. The authors provide a theoretical analysis of the error bounds of Stable-Sketch, which shows that it achieves high accuracy with a small memory footprint.

3. The paper also provides an empirical evaluation of Stable-Sketch that demonstrates its superior performance compared to existing state-of-the-art algorithms.


The main technical novelty of the paper lies in the utilization of multi-dimensional statistics over existing uni-dimensional ones for replacement decisions. More specifically, the paper uses bucket stability in addition to item information for deciding eviction probability. Empirical results are quite impressive, especially in the low memory regime compared to the state of the art.

**Questions:**

1. In section 3.1, the authors mention that they achieve memory efficiency by avoiding repetitive hash operations. However, this should potentially affect the accuracy of the model. How does it then achieve both better accuracy and memory efficiency compared to existing methods?

2. What's the effect of the threshold parameter for the heavy hitter problem on memory and other constraints?

**Reviewer Confidence:**

2: The reviewer is willing to defend the evaluation, but it is likely that the reviewer did not understand parts of the paper

**Scope:**

4: The work is relevant to the Web and to the track, and is of broad interest to the community

---

### Official Review · Reviewer_qy5j · 2023-11-22

**Novelty:** 5
**Technical Quality:** 6

**Review:**

Sketches are widely used and studied in data stream processing. This paper introduces a novel statistic called “bucket stability”, and incorporates it with other statistics to build the sketch on a data stream. For Sketch, this newly added statistic is lightweight in terms of computation and memory footprint.

Strengths:

S1: The paper is well organized and easy to follow.

S2: The problem is well defined. The “bucket stability” concept is novel.

S3: The paper presents a sound theoretical analysis about the error bound of the sketch.

Weaknesses:

W1: Although many methods were compared in the experiment, the latest work compared was from 2021. There are also some newer methods that can also participate in the comparison. For example, the references listed below, such as [1], [2], etc.

[1] Q. Xiao, X. Cai, Y. Qin, Z. Tang, S. Chen and Y. Liu, "Universal and Accurate Sketch for Estimating Heavy Hitters and Moments in Data Streams," in IEEE/ACM Transactions on Networking, vol. 31, no. 5, pp. 1919-1934, Oct. 2023, doi: 10.1109/TNET.2022.3216025.

[2] Francesco Da Dalt, Simon Scherrer, and Adrian Perrig. 2022. Bayesian Sketches for Volume Estimation in Data Streams. Proc. VLDB Endow. 16, 4 (December 2022), 657–669. https://doi.org/10.14778/3574245.3574252

W2: Regarding Algorithm 1, the update strategy for the stability statistic (B(R, M).S) needs more explanation to readers. Although it seems reasonable, it is best to explain why it is designed this way.

W3: The data sets used in the experiments are all network traffic data. Is there any web stream data or text stream data can be used to evaluate the performance of the method proposed in this paper? In particular, in web streams, keys (e.g., urls) are generally longer. In this case, perhaps the Stable-Sketch* method mentioned in the paper can play a greater role. This may bring new challenges.

W4: In the SIMD optimization discussed in Section 5.4, the key matching process is optimized to use SIMD parallel processing. This is not entirely consistent with Algorithm 1. Will this cause new problems?

**Questions:**

W1, W3

**Reviewer Confidence:**

4: The reviewer is certain that the evaluation is correct and very familiar with the relevant literature

**Scope:**

3: The work is somewhat relevant to the Web and to the track, and is of narrow interest to a sub-community

---

### Official Review · Reviewer_3Lhp · 2023-11-23

**Novelty:** 5
**Technical Quality:** 6

**Review:**

The paper presents a sketch, Stable-Sketch, similar to the CountMin sketch, for queries on data streams that allow tasks that are more difficult than usual such as heavy hitters and changers. The sketch is based on the notion of bucket stability, which tracks how often items are changed in a bucket.

Some theoretical bounds are given on the estimation errors and the error bound of the Stable-Sketch.

The experimental study is performed on datasets, mostly from the networking domain, by comparing to different baselines for several metrics, by keeping the same memory budget. The results show that the Stable-Sketch is more accurate than the baselines for the tasks of heavy hitters and persistent item detection. The interesting fact is that the implementation was also performed on a switch, using the P4 programming language.

Generally, the paper is well written and the approach seems convincing.
On the other hand, the paper is not self contained. Multiple references are made to the appendix (section 4 and the practical implementation); these should be integrated into the main paper if possible. This issue is especially important in Section 5.5, which refers exclusively to the appendix.

Some other issues are detailed in the question section.

**Questions:**

1. In Section 4, the estimation error and the error bounds are given as-is. First, they are not self-contained, as detailed above, and second, they are not explained. The paper should explained how this related to the previous related work, and also explained what the results mean in practice.

2. There may be a lack of understanding on my part, but it is not at clear to me whether this sketch is good *in general* or only for the tasks that are evaluated in the paper. What I mean: is the sketch *worse* for other, more general tasks? Maybe this should be clarified.

**Reviewer Confidence:**

3: The reviewer is confident but not certain that the evaluation is correct

**Scope:**

3: The work is somewhat relevant to the Web and to the track, and is of narrow interest to a sub-community

---

### Official Review · Reviewer_w4cS · 2023-11-27

**Novelty:** 6
**Technical Quality:** 5

**Review:**

## Quality
The quality of the work is high, as evidenced by:
1. Comprehensive Theoretical Analysis: The paper provides a detailed theoretical analysis of the error bounds of Stable-Sketch, offering credibility to its claims.
1. Extensive Experimental Evaluation: The authors conducted a wide range of experiments using real-world datasets, which helps validate the effectiveness of Stable-Sketch in practical scenarios.
1. Clear Methodology: The approach to the problem and the development of the solution is methodically explained, showing a well-structured research process.

## Clarity
The paper is well-written and structured, making it accessible:
1. Well-Organized Sections: The paper effectively introduces the problem, discusses related work, presents the Stable-Sketch framework, and then delves into experimental results.
1. Clear Explanations: Technical concepts are explained in a manner that is understandable, with a balance between technical depth and readability.
1. Effective Visuals: The use of figures and tables enhances understanding, especially in conveying experimental results.

## Originality
The paper demonstrates originality in its approach:
1. Novel Concept of Bucket Stability: Introducing the idea of bucket stability as a metric for item variation is a novel approach in the context of data stream processing.
1. Unique Combination of Techniques: The combination of multidimensional information and a stochastic approach for item replacement in sketches is a unique contribution to the field.

## Significance
The work is significant due to:
1. Addressing Practical Limitations: The paper tackles real-world limitations like memory constraints and high-speed data stream processing.
1. Applicability in Diverse Fields: The application of Stable-Sketch in various web-centric areas, including anomaly detection and recommendation systems, demonstrates its broad relevance.
1. Open Source Contribution: Making the source code available on GitHub encourages further research and practical application, enhancing the paper’s impact.

Pros
1. Innovative Approach: Leveraging bucket stability for data stream processing is a significant innovation.
1. Robust Experimental Validation: The use of diverse datasets and thorough experimental setups validate the claims.
1. Practical Viability: The paper not only focuses on theoretical aspects but also demonstrates the practical implementation and potential real-world impact.
1. High Detection Accuracy and Speed: Demonstrated ability to achieve high accuracy and processing speed, even under memory constraints.

Cons
1. Complexity for Practical Deployment: While the paper discusses practical deployment, the complexity of the Stable-Sketch algorithm might pose challenges in certain real-world scenarios.
1. Limited Discussion on Limitations: The paper could benefit from a more detailed discussion on potential limitations or scenarios where Stable-Sketch might not perform optimally.
1. Comparative Analysis: While there is a comparison with existing methods, a more detailed discussion on how Stable-Sketch performs relative to the state-of-the-art in various scenarios could be beneficial.

**Questions:**

1. Could you elaborate on how Stable-Sketch performs in various computing environments with different hardware constraints, especially in lower-end systems or systems with different cache architectures?
1. How does the performance of Stable-Sketch scale with increasingly large datasets and high-speed data streams?
1. How effective is Stable-Sketch in environments with highly dynamic and unpredictable data streams? Is there a performance trade-off in such scenarios?

**Reviewer Confidence:**

3: The reviewer is confident but not certain that the evaluation is correct

**Scope:**

4: The work is relevant to the Web and to the track, and is of broad interest to the community

---

### Official Review · Reviewer_66uH · 2023-11-29

**Novelty:** 6
**Technical Quality:** 7

**Review:**

Pros:
1. The introduction of a multi-dimensional, information-rich data sketch that can adapt to variability in data streams. Especially the concept of bucket stability, which could significantly improve detection accuracy.
2. Emphasize both theoretical and practical aspects, including source code availability, and also provide mathematical analysis on the error bound of this new method.
3. Addressing both memory constraints and detection accuracy, which are often at odds in data stream processing.
4. The evaluation section provides comprehensive testing and comparisons with existing systems.
Cons:
1. The paper's technical depth might limit its accessibility to a broader audience.

Overall, the work presented in the paper is a substantial contribution to the field of data stream processing.

**Questions:**

Are there potential limitations or scenarios where Stable-Sketch may not perform as expected?

**Reviewer Confidence:**

3: The reviewer is confident but not certain that the evaluation is correct

**Scope:**

4: The work is relevant to the Web and to the track, and is of broad interest to the community

---

### Decision · Program_Chairs · 2024-01-22

**Decision:**

Accept (Oral)

**Comment:**

The paper proposes an approach called Stable-Sketch to enable querying data streams with tasks that are more difficult than than the often-used heavy hitters and changers. The sketch is based on the novel notion of bucket stability, which tracks how often items are changed in a bucket.

 The reviewers unanimously agree that the work is of high-quality in terms of novelty, theoretical soundness and rigor, as well as extensive experiments considering multiple datasets and baselines. I recommend an accept, and encourage the authors to address minor reviewer concerns before camera ready, e.g. making the paper more self-contained.